# STRUCTURE-PRESERVING TEXT-BASED EDITING FOR FEW-STEP DIFFUSION MODELS

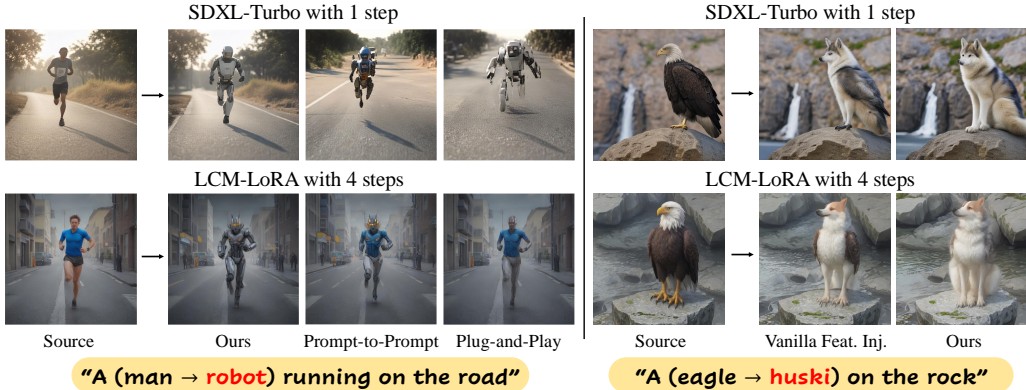

Figure 1: Text-based image editing results with few-step DMs. Previous methods (injecting the source attention map) validated on multi-step diffusion models fail to preserve the source image structure or tend to neglect the target prompt. We observe injecting the feature instead of attention leads to the transfer of structure, but vanilla feature injection fails to remove the source appearance. Our method effectively preserves the source structure while avoiding the attribute leakage issue.

## ABSTRACT

Text-based image editing aims to generate an image that corresponds to the given text prompt, but with the structure of the original source image. Existing methods often rely on attention maps in diffusion models (DMs) for structure preservation, as these features are considered to play a primary role in determining the spatial layout. However, we find that these methods struggle to preserve the spatial layout when applied to few-step DMs (e.g., SDXL-Turbo), limiting their use cases to the slower multi-step DMs (e.g., Stable Diffusion). In this work, we investigate the limitations of these approaches in terms of intermediate feature representations. Our findings indicate that for few-step DMs, the attention layers have less influence in determining the structure. To tackle this, we localize layers within the network that better control spatial layout and inject these features during the editing process. Additionally, we disentangle structural information from other features to avoid conflicts between the injected features and the text prompt. This ensures that the edited image faithfully follows the prompt while preserving the source structure. Our method outperforms existing text-based editing baselines.

## 1 INTRODUCTION

Recent advances in generative modeling have led to the development of text-to-image (T2I) Diffusion Models (DMs) (Rombach et al., 2022; Saharia et al., 2022; Betker et al., 2023; Podell et al., 2023; Esser et al., 2024), which generate high-quality images from text prompts. However, the generation process is computationally intensive and time-consuming due to the many reverse diffusion steps required. To address this, recent approaches such as the Latent Consistency Model (LCM) (Luo et al., 2023a) and ADD (Turbo) (Sauer et al., 2023) have been proposed, enabling image generation with significantly fewer steps (e.g. less than 4 steps).

Meanwhile, the release of off-the-shelf large-scale T2I DMs has brought increased interest in text-based image editing methods (Hertz et al.; Tumanyan et al., 2023; Cao et al., 2023). Here, this editing

task focuses on two fundamental objectives: 1) the edited image should align with the provided text-prompt, and 2) attributes not explicitly specified in the prompt should be preserved with that of the original source image. In particular, preserving the structural information of the original source image, such as the overall spatial layout and object poses, is crucial in this task. To accomplish this, many editing methods (Hertz et al.; Tumanyan et al., 2023; Cao et al., 2023; Chung et al., 2024; Hertz et al., 2024) utilize the fact that attention maps in self- and cross-attention layers significantly affect the spatial layout of synthesized images, and provide them as a structural guidance during the editing process. Within this paper, we refer to these methods as *attention injection*.

However, we observe that existing text-based image editing techniques validated on multi-step DMs face challenges when applied to few-step DMs, such as SDXL-Turbo (Sauer et al., 2023; Podell et al., 2023) and LCM-LoRA (Luo et al., 2023b). As shown in Fig. 1, prior attention-based methods (e.g. P2P (Hertz et al.) and PnP (Tumanyan et al., 2023)) often fail to produce satisfactory results when used with these models. They either fail to preserve the spatial layout of the source image or result in degraded image quality.

To investigate the causes of these failures, we analyze how attention map injection affects the features in few-step DMs as in Fig. 2. A notable difference against multi-step DMs is that attention maps in few-step DMs do not trigger meaningful structural changes, as visualized in Fig. 2.

We hypothesize that in few-step DMs, the spatial layout is mainly determined by features in the non-residual [1] path, rather than self- and cross-attention maps that exist in residual form (i.e., will be added to the skip connection), as in Fig. 2. This motivates us to modify the features in the non-residual path, specifically those between generator blocks without skip connections. This way, we provide explicit structural guidance to the output image, that can not be ignored via skip-connection. In Fig.2, we verify that this type of approach, which we refer to as *feature injection*, can successfully generate images that share the overall structure with that of the source image, as desired.

However, feature injection has its own drawback: attribute leakage. In the second-rightmost image of Fig. 4-(b), the target attribute (huski) exhibits specific attributes of the source image (eagle). This indicates that the appearance and structural information are entangled within the injected feature. Thus, by simply substituting the entire feature as is, the appearance information of the source image may leak, causing conflicts with the edit prompt that specifies contradictory attributes.

Accordingly, we aim to design a feature that provides structural guidance without interfering with the edit prompt. To extract structure-relevant information from features, we apply Singular Value Decomposition (SVD) to the source image feature by treating each spatial feature as a data point. We observe that the components with the highest variance across the spatial dimension mainly capture structural information rather than appearance, as spatial information tends to vary most across pixels in an image. Thus, we leverage these high-variance directions from the source image, combining them with appearance information (e.g., identity, object class) from the edited features. This way, we obtain the edited feature that has the structural information of the source image, but with the overall appearance information of target edited image, aligning with the objective of the text-based image editing task.

However, simply combining features leads to undesirable artifacts due to the spatial misalignment between the source-structure and edit-appearance features. To address this, we propose a spatial matching technique to further improve the editing quality. In detail, we compute a correspondence map between features of the source and the edited image, and warp the edited appearance features accordingly. At last, we demonstrate that the proposed method outperforms baseline methods in both qualitative and quantitative comparisons for text-based editing in the few-step DMs.

## 2 RELATED WORKS

**Few-step Diffusion Models.** Despite the impressive generative capabilities of DMs, their practical useage is limited by the time-consuming iterative denoising steps in the reverse process. Numerous works (Ho et al., 2020; Song et al.; Meng et al., 2023) aim to reduce computational complexity by fine-tuning DMs or decreasing the number of denoising steps. For exmaple, LDM (Rombach et al.,

---

[1]The term *non-residual* and *skip connection* will be used interchangeably. These indicate the identity path, in terms of ResNet Blocks, as shown in Fig. 2-(a). Note that the term skip-connection here does not indicate the UNet long-term skip connections from encoders to decoders.

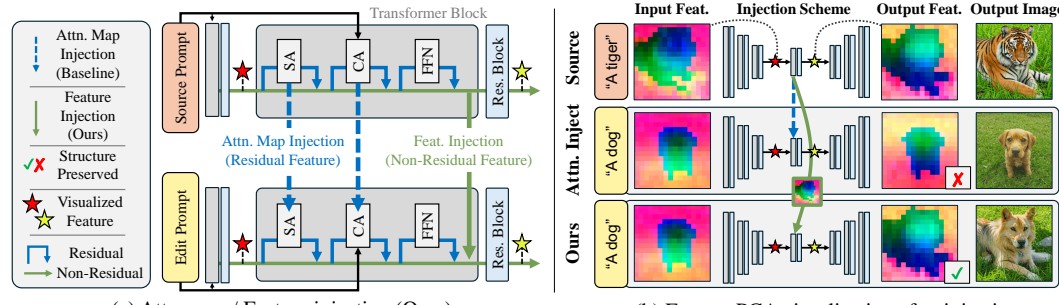

(a) Attn. map / Feature injection (Ours)                 (b) Feature PCA visualization after injection

Figure 2: **(a)** Conceptual comparison between attention (-map) injection and feature injection. Attention injection mainly utilizes cross-attention (CA) and self-attention (SA) maps, which are in the residual path, to transfer the structure of the source image. In contrarily, the non-residual feature serves as the key factor for feature injection. **(b)** Visual examples of SDXL-Turbo features, before and after applying each injection method. Attention-injection fails to preserve the source structure, while feature injection successfully generates an image with the source image's structure.

2022) reduces generation time by denoising in a perceptually compressed latent space. Moreover, several works have focused on reducing the number of timesteps. For example, LCMs (Song et al., 2023; Luo et al., 2023a) and ADDs (Sauer et al., 2024; 2023) focus on distilling pretrained latent DMs and demonstrate impressive performance with steps fewer than four.

Despite these efforts to reduce timesteps, most editing works still rely on tens or hundreds of iterative denoising steps. In this work, we improve the applicability of editing methods on few-step DMs.

**Text-based Image Editing.** Recent advances in text-to-image generative models (Gafni et al., 2022; Ramesh et al., 2022; Rombach et al., 2022; Esser et al., 2024) have shown significant progress in generating high-quality, diverse images. Building on this, several works (Ruiz et al., 2023; Gal et al., 2022; Tumanyan et al., 2023; Hertz et al.) have introduced novel attention-based methods for text-based image editing. For instance, Prompt-to-Prompt (Hertz et al.) manipulates cross-attention layers for control over spatial alignment between image content and text, while Plug-and-Play (Tumanyan et al., 2023) reveals that self-attention maps guide the structural attribute of generated images. However, these methods, which rely on multi-step DMs, are hindered by the inefficiencies introduced through multiple denoising steps.

To address these limitations, recent approaches (Deutch et al., 2024; Wu et al., 2024) have explored text-based editing within few-step DMs. These methods mainly improve the inversion process and enhance text-image alignment through techniques such as large-scale language models (LLM) (Liu et al., 2024; Ouyang et al., 2022) or Score Distillation Sampling (SDS) (Poole et al.). Unlike their multi-step counterparts, these methods focus on the inverted noise space while overlooking the intermediate feature space of the model. Differently, in terms of intermediate feature representation, we propose an editing approach that can successfully provide structural guidance, while also retaining the ability to reflect the text prompt.

## 3 METHOD

### 3.1 ATTENTION INJECTION AND FEATURE INJECTION IN FEW-STEP DIFFUSION

**Limitation of Attention Injection in Few-step DMs.** The most representative works, such as P2P (Hertz et al.) and PnP (Tumanyan et al., 2023), uncover the fact that attention maps have a significant impact on the spatial layout of generated images, and thus, utilize them to preserve the structure within the editing process. In this paper, we refer to these kinds of methods of replacing the attention maps as attention injection.

A straightforward approach to perform text-based editing with few-step diffusion models (DMs) is to simply apply attention injection on them. As a preliminary experiment, we apply attention injection on SDXL-Turbo in Fig. 2-(b). Here, we observe that the effects of the attention layers in the few-step model are significantly lower than their role in multi-step DMs, when determining the spatial layout. Contrary to conventional multi-step DMs (e.g., Stable Diffusion), we cannot manipulate

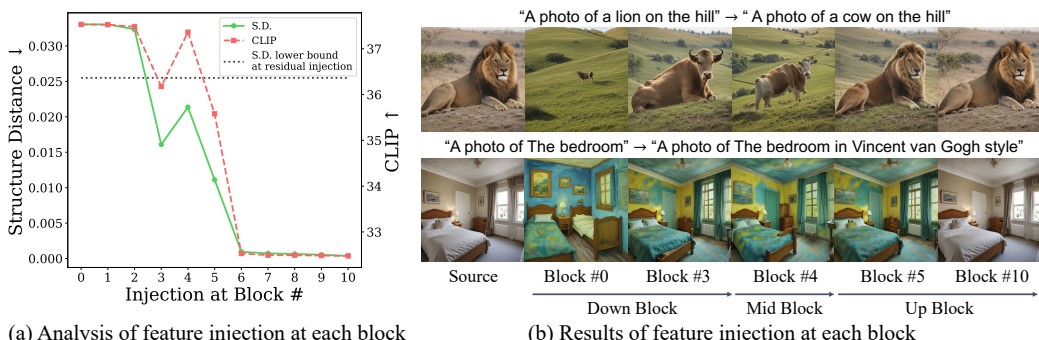

(a) Analysis of feature injection at each block     (b) Results of feature injection at each block

Figure 3: Analysis on attention and feature injection results, while varying the injection location. **(a)** Structure distance between the source and the attention- or feature-injected images and CLIP similarity between the source image and the text condition. Attention injection exhibits a significantly smaller degree of impact in reducing the structural distance (dashed horizontal line). Feature injection at all blocks, except for Block #3, fails to either maintain the source structure (S.D.), or to generate an image aligned with the text prompt (CLIP). **(b)** Visualization for the identical experiment. Results align with the quantitative analysis, where only Block #3 edits the image as desired.

the spatial layout to follow the structure of the source image by simply replacing attention maps. Considering the phenomenon above, we hypothesize that this failure is due to the characteristics of few-step DMs, where the output features of the self-attention (SA) and cross-attention (CA) blocks have less contributions in determining the spatial layout. Since these features are part of residual connections (i.e., will be added to the identity value), it may be easier to ignore them when determining the spatial layout. This claim is supported by our analysis in Fig.3-(a) where attention injection fails to sufficiently reduce the structural distance, regardless of the injection location. Notably, we emphasize that this failure occurs regardless of the model architecture, SDXL-Turbo and Latent Consistency Model (LCM).

**Analyzing Feature Injection in Few-step DMs.** Thus, we aim to modify the features residing in the non-residual path directly (Fig. 2), specifically, the features between the generator blocks. This is in order to perform feature manipulation in a manner that cannot be ignored via skip-connection. In Fig. 3, since each block represents different levels of information, we first analyze[2] the effect when the edit-feature of a specific block is substituted by the source feature from the identical block.

To explore the effects of each individual building block, we apply feature injection on each block and evaluate its suitability for text-based editing by measuring 1) the structural distance between the edited image and source image and 2) the CLIP similarity between the text condition and edited image. Here, the structural distance is computed by the L2 distance of the self-similarity maps of each image, based on DINOv2 (Oquab et al., 2023). Note that the self-similarity of DINOv2 is known to effectively represent the structural distance (Tumanyan et al., 2022). The two metrics above (structural distance, CLIP similarity) reflect the effectiveness of the edited images in preserving the source image's structure and adhering to the text condition, which are crucial for text-based image editing. As shown in Fig. 3-(a), we observe that feature injection in the too earlier and later blocks (i.e. Block #0-2, #6-10) either fails to maintain the structure of source, or neglects given text condition. Thus, we consider intermediate layers (Block #3-5) as the candidates for feature injection.

To build further intuition on the impact of feature injection on each block, we visualize sythesized results in Fig. 3-(b). Aligning with Fig. 1-(a), we observe that the source structure preservation is achieved when feature injection is applied on the Block #3 and latter layers. However, injection of the Block #5 and latter layers affect in an undesirable manner, where the edited image resembles certain aspects of the source image attributes, more than required. Additionally, while the application on Block #4 does capture the coarse spatial layout, it often fails to preserve the detailed structure. This is shown in both the edit results (the cow image, smaller than required; and the bedroom image with wrong window size) in Fig. 3-(b) at Block #4, and the abrupt peak in Fig. 3-(a) at Block #4.

---

[2]Analysis was conducted on the Imagen dataset (Zou et al., 2024) with 360 pairs of text prompts.

Meanwhile, we observed that by applying feature injection on a well-chosen layer (Block #3 in SDXL-Turbo) among the candidate blocks, we can constraint the synthesized image structure to align with the source images, while accurately representing the text prompt. To further validate the generalizability, we perform identical experiments on another representative few-step model, LCM. We observe similar effects when applying injection at the output of Block #5. Interstingly, these two locations to inject are the output of the last encoder layer, which is used as a skip-connection feature for the first decoder block. We conjecture that these features have an appropriate level of structural information, since they are the (UNet long-term-) skip-connection of coarsest resolution.

In summary, these experiments suggest that applying feature injection to an appropriate layer enables effective text-based image editing, and this approach is expected to work for diverse editing circumstances generally. Accordingly, we build our method upon this observation. In the following sections, we will further analyze and improve this approach to enhance its effectiveness. For simplicity, further details will be described based on SDXL-Turbo.

## 3.2 DECOMPOSING STRUCTURAL AND APPEARANCE FACTORS IN FEW-STEP DMS

**Entangled Structure and Appearance Factors within the Feature Space.** Above, we observed that features extracted from specific blocks (e.g. Block #3) encode spatial information. However, while these features effectively capture spatial layouts, they still retain some appearance attributes from the source image which may potentially hinder the overall editing process. For example, examining the rightmost image in Fig. 4-(b) (marked as '∼256'), we observe that directly injecting these features leads to suboptimal editing. The edited image maintains the appearance of the source image (narrow eyes and short ears), which we indicate as the attribute leakage phenomenon. This occurs because the tiger's appearance information exists in the injected features (i.e. Block #3 features), and interferes with the synthesis process. This observation emphasizes the necessity of a method to isolate only the structural information before injecting into the editing pipeline.

**Structural Information in Top-K Singular Components.** To gain intuitions. we apply Singular Value Decomposition (SVD) to features obtained from a single image, by treating each spatial feature as a data point. The intuition behind it is that the features already contain abundant and diverse spatial information, so only the basis with the highest variance would mostly contain structure-relevant information rather than appearance. Also, the most variable information (i.e. directions with the highest variances) across pixels in a single image would be spatial information, as appearance information such as textures of styles can be shared across most pixels. In the first row of Fig. 4-(b), we visualize the image feature projected on the top significant singular vectors (#0-3), an also those projected onto less significant singular basis (#15-18). The results from the projections onto the top singular vectors clearly reflects spatial layout of each image. In contrast, projections onto the lower singular vectors (#15–18) reveal almost no structural information, which aligns with our intuition.

To further validate and quantify the statements about the decomposition between structure-aware information and appearance information, we conducted further analysis. We analyze the correlation between singular vectors associated with large singular values and the spatial information. In detail, we gradually project the features on basis vectors from highest to lowest singular values and accumulate them. Then, we quantify the structural information that the projected features have. To quantify the structural information, we measure the structural distance between the projected features and that of DINOv2 feature extracted from the generated images. Since the similarity map from DINOv2 is known to represent the structure of image (Tumanyan et al., 2022), this distance can be interpreted as a measurement regarding the degree of involvement of structural information in the generator feature. As observed in Fig. 4-(a), similarity maps' differences rapidly decrease when singular values are high. Differently, the error between the projected and original features is more smoothly reduced, supporting the claim that components of larger singular values mostly contain spatial information instead of appearance-relevant ones.

**Analysis on Non Top-K Components.** Regarding the linearity of SVD, information apart from the structure can be simply obtained by projecting a feature onto the non top-K components. This information captures attributes other than shape, such as object classes or style, which we will refer to as *appearance information*. This claim is supported by the second row of Fig. 4-(b). We observe that transferring components with high singular values preserves the overall structure of the source image, while low singular vectors maintain the detailed appearance. With these insights, the following

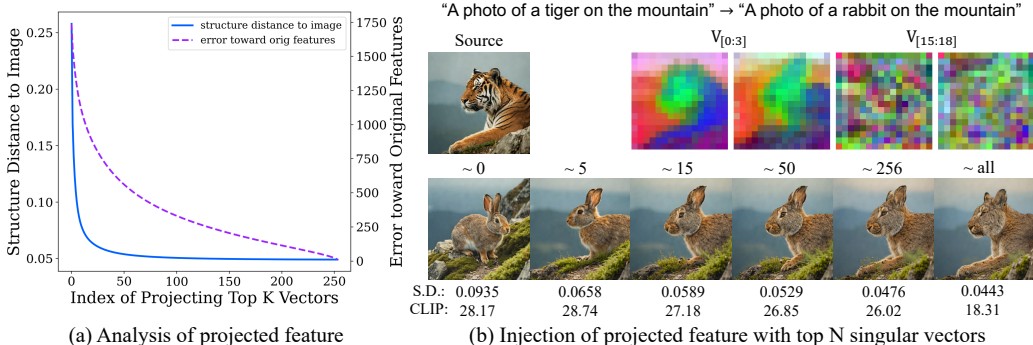

(a) Analysis of projected feature      (b) Injection of projected feature with top N singular vectors

Figure 4: Analysis on feature injection, while varying the singular value index threshold. We observe that the basis of high singular values contains structural information. (a) Structure distance between the projected features and DINOv2 features from synthesized images, measured by the distance between self-similarity maps of each. The structure distance rapidly decreases when projecting on a few top singular vectors, while error toward the original features largely remains. Therefore, it implies that spatial information of images mostly resides in the high singular vectors. (b) Feature injection results with source features projected on the top-K basis vectors. As K increases, the editing results resemble the more detailed structure of the source image, but also the appearance (attribute leakage). We also visualize the features projected by the top 0-3 and 15-17 basis ($\mathbf{V}_{[0:3]}$ and $\mathbf{V}_{[15:18]}$), which reflect the dominance of structural information in the high singular values.

section describes a method that decomposes the features into structure and appearance components and substitutes only the structure-relevant features from the source into the edit-features.

### 3.3 STRUCTURE-PRESERVING TEXT-BASED EDITING (SPEDIT)

**Overview.** Based on the observations above, we develop our main method with the following three key ideas: 1) disentangling the structural and the appearance information in the feature representation by SVD and projection, 2) improving spatial alignment between the source-structure and edit-appearance, and finally 3) compositing these features and performing feature injection with it. This way, we provide structural guidance without harming the editing quality.

**Structure-Appearance Decomposition** Here, we elaborate on the details to isolate the structural information from the source feature, and the appearance information of the edit feature. To reemphasize, this decomposition is designed in order to follow the given text condition, while preserving the source image structure.

First, we start with the noise identical to the source image for synthesizing the edited image in the reverse diffusion process. Also, for the sake of simplicity, we omit the layer index and timestep notation. Let the features of the source and edited images during the diffusion reverse process be denoted as $F_{\text{src}} \in \mathbb{R}^{hw \times c}$ and $F_{\text{edit}} \in \mathbb{R}^{hw \times c}$, respectively. To decompose the spatial information from $F_{\text{src}}$, we first apply SVD to $F_{\text{src}}$ and obtain the basis of channel vectors $\mathbf{V}$. As mentioned earlier, spatial information is dominant in the components of high singular values. Thus, we project the source feature $F_{\text{src}}$ by top-K basis vectors $\mathbf{V}_{[0:k]}$ to decompose the spatial information from it.

Then, we also project the feature of edited image $F_{\text{edit}}$ by remaining basis vectors $\mathbf{V}_{[k:]}$ to maintain the appearance of the target image. This process is defined as follows:

$$F_{\text{src}}^s = F_{\text{src}}\mathbf{V}_{[0:k]}^\top, \quad F_{\text{edit}}^a = F_{\text{edit}}\mathbf{V}_{[k:]}^\top, \tag{1}$$

where $F_{(\cdot)}^s$ and $F_{(\cdot)}^a$ are the structural and appearance features, and $[\cdot, \cdot]$ is the concatenation operation, and $\mathbf{V}_{[\cdot:\cdot]}$ are basis vectors of a specific index range.

**Spatial Matching with a Correspondence Map.** Since spatial information may still remain in the projected features of the edited image $F_{\text{edit}}^a$, potential conflicts may arise leading to degraded visual quality. Thus, we additionally modify the edit-appearance feature to have the spatial layout of the source image. To this end, we warp the appearance feature, by computing a correspondence map between the source and edited features.

Figure 5: Conceptual illustration of our method. We modify and substitute the edit feature during the reverse diffusion step initiated by the edit-text prompt. This modified feature contains the appearance information of the edit feature, but the structural information of the source feature. This modification is performed by concatenating i) the source feature projected on the top-K significant singular vectors (structure) and, ii) the edit feature projected on the non top-K singular vectors (appearance). Within this process, we further facilitate the spatial alignment of the appearance information between the source and edit feature, based on the spatial correspondence matching scheme.

Thanks to the emergent property of diffusion features for dense correspondence (Tang et al., 2023; Zhang et al., 2024), we obtain this map by computing the cosine similarity between source and edited features ($F_{\text{src}}, F_{\text{edit}}$). Specifically, we utilize the structural features ($F_{\text{src}}^s, F_{\text{edit}}^s$) projected by top-K components from SVD for only considering the structural similarity. The correspondence map $W \in \mathbb{R}^{hw \times hw}$ is then computed as the pair-wise cosine similarity between $F_{\text{src}}^s$ and $F_{\text{edit}}^s$. Once $W$ is obtained, we simply warp the edited appearance feature $F_{\text{edit}}^a$ using $W$, aligning the features for editing as in Eq. 1. This process is defined as follows:

$$W^{(i,j)} = \text{cosim}(F_{\text{src}}^{s\,(i)}, F_{\text{edit}}^{s\,(j)}), \tag{2}$$

$$\hat{F}_{\text{edit}}^a = \text{softmax}(W F_{\text{edit}}^a \times \tau), \tag{3}$$

where $\text{cosim}(\cdot, \cdot)$ is a cosine similarity between the inputs and $\tau$ is the temperature scaling parameter for sharpening the correspondence map $W$.

**Structure-aware Feature Injection.** Finally, we concatenate the projected structure feature of the source image $F_{\text{src}}^s$ and the warped appearance feature of the target prompt $\hat{F}_{\text{edit}}^a$, as below:

$$\hat{F}_{\text{edit}} = [F_{\text{src}}^s, \hat{F}_{\text{edit}}^a]\, \mathbf{V}. \tag{4}$$

Then, these concatenated features are injected during the reverse diffusion process by replacing the features of edited image $F_{\text{edit}}$ with the concatenated feature $\hat{F}_{\text{edit}}$.

## 4 EXPERIMENTS

### 4.1 EXPERIMENTAL SETUP

**Benchmark Datasets.** We validate our proposed method on two comprehensive benchmark datasets for text-based image editing: Generated ImageNet-R-TI2I (Tumanyan et al., 2023) and PIE-Bench (Ju et al., 2024). Specifically, Generated ImageNet-R-TI2I comprises 146 pairs of generated source images and target prompts, while PIE-Bench consists of 700 pairs of real source images and target prompts with 10 editing types.

**Evaluation Metrics.** For evaluation, we employ two commonly used metrics: CLIP score and Structure Distance (Tumanyan et al., 2023). The CLIP score measures the cosine similarity between CLIP (Radford et al., 2021) features of the edited images and the corresponding target prompts, reflecting how well the edits align with the given prompts. Structure Distance is computed as the L2 difference between DINO self-similarity maps of the source and edited images, quantifying the degree of structure preservation. For both metrics, we use DINOv2 (Oquab et al., 2023; Caron et al., 2021) feature extractor with ViT-B-14 (Dosovitskiy, 2020).

**Implementation Details.** We compare our proposed method with the default settings of the baselines' code of the public repository. Any modifications are discussed in the corresponding sections. All computation times are measured on a single NVIDIA A6000 GPU. For other hyperparamters, we

| Method | Model | Generated ImageNet-R-TI2I | | | |
| --- | --- | --- | --- | --- | --- |
| | | CLIP $\uparrow$ | S.D. ($\times 10^3$) $\downarrow$ | Sampling Steps $\downarrow$ | Time (sec) $\downarrow$ |
| P2P | SD v1.5 | 24.72 | 21.15 | 50 | 14.87 |
| PnP | SD v1.5 | 27.37 | 35.82 | 50 | 11.19 |
| P2P | LCM-LoRA | 20.56 | 7.52 | 4 | 0.45 |
| PnP | LCM-LoRA | 20.44 | 5.95 | 4 | 0.42 |
| **Ours** | LCM-LoRA | **20.80** | **5.57** | 4 | 0.40 |
| P2P | SDXL-Turbo | 26.05 | 29.14 | 1 | 0.49 |
| PnP | SDXL-Turbo | 26.10 | 27.58 | 1 | 0.40 |
| **Ours** | SDXL-Turbo | **26.19** | **27.46** | 1 | 0.42 |
| P2P | SDXL-Turbo | 26.04 | 31.73 | 4 | 1.06 |
| PnP | SDXL-Turbo | 26.99 | 35.43 | 4 | 0.73 |
| **Ours** | SDXL-Turbo | **27.34** | **31.21** | 4 | 0.93 |

Table 1: Comparison on Generated ImageNet-R-TI2I dataset. S.D. denotes Structural Distance.

use (Block #3, $k = 60$, $\tau = 50$) and (Block #5, $k = 15$, $\tau = 15$), for experiments on SDXL-Turbo and LCM-LoRA, respectively.

## 4.2 QUANTITATIVE RESULTS

**Comparison to baselines on generated images.** For evaluation, we employ two few-step DMs, SDXL-Turbo (Podell et al., 2023; Sauer et al., 2023) and LCM-LoRA (Luo et al., 2023b), to confirm the robustness of the proposed method. We mainly compare ours with editing techniques built upon in many-step DMs (Prompt-to-Prompt (P2P) (Hertz et al.) and Plug-and-Play (PnP) (Tumanyan et al., 2023)), but on top of few-step models. As presented in Tab. 1, our method consistently outperforms both P2P and PnP across CLIP and Structure Distance metrics regardless of the model type. These results demonstrate the consistent and superior editing performance of our approach in few-step DMs. When comparing ours on the few-step SDXL-Turbo (5[th] row) against baseline methods on the many-step SD v1.5 (1[st]-2[nd] row); we achieve comparable performance, while requiring a order of degree fewer steps (4 / 50) for synthesizing.

**Comparison to baselines on real images.** In Tab. 2, we validate our proposed method for the real-world image editing task. Since this requires an inversion step, we employ TurboEdit[2] (Deutch et al., 2024) and apply our method during the denoising steps. To evaluate P2P and PnP, we use DDIM-inversion (Song et al.) and EF (Huberman-Spiegelglas et al., 2024) to invert images on SD v1.5. Additionally, since EF, TurboEdit[1] (Wu et al., 2024) and TurboEdit[2] (Deutch et al., 2024) can work as a standalone editing method, we additionally report editing results of them, as is.

In the few-step configuration (4 steps), the proposed method significantly improves structure distance against TurboEdit[2], with a comparable CLIP score. Notably, this experimental result implies that our approach preserves structural information from the source image while effectively incorporating text-based edits. Also, we compare our method with TurboEdit[1], the concurrent editing technique in SDXL-Turbo, and validate we achieve a higher CLIP score. Due to the lack of public availability for TurboEdit[1], we only report the CLIP score from the original paper.

When compared to multi-step baselines (1[st]-4[th] row), the proposed method achieves a slightly lower CLIP score, while reaching a significantly better structure distance. This indicates that the baseline methods lose the structural information of the source image and heavily rely on the target prompt for generation. In contrast, our method shows a better trade-off between structural preservation and text fidelity. Overall, our approach yields superior results across both few-step and many-step diffusion models when considering both metrics.

## 4.3 QUALITATIVE RESULTS

**Comparison to baselines on generated images.** For a qualitative comparison, we present the edited samples in Fig. 6, using the same settings as in Tab. 1. Our edited samples show better structure preservation while successfully incorporating the target prompts. For instance, as shown in the first column of Fig. 6, P2P generates three birds, and PnP produces only one, but our method effectively generates both a flower and a bird, while altering the style to "cartoon". These results highlight that

| Method | Model | PIE-Bench | | | | |
|---|---|---|---|---|---|---|
| | | CLIP ↑ | S.D. (×10³) ↓ | Inversion Steps ↓ | Sampling Steps ↓ | Time (sec) ↓ |
| DDIM + P2P | SD v1.5 | 25.11 | 32.54 | 50 | 50 | 14.87 |
| DDIM + PnP | SD v1.5 | 25.41 | 18.48 | 50 | 50 | 11.19 |
| EF | SD v1.5 | 27.06 | 18.84 | 50 | 50 | 6.11 |
| EF + P2P | SD v1.5 | 26.99 | 35.43 | 50 | 50 | 10.84 |
| TurboEdit[1] | SDXL-Turbo | 25.05 | − | 4 | 4 | — |
| TurboEdit[2] | SDXL-Turbo | **26.52** | 19.07 | 4 | 4 | 1.52 |
| TurboEdit[2] + **Ours** | SDXL-Turbo | 25.57 | **13.10** | 4 | 4 | 1.57 |

Table 2: Comparison on PIE-Bench dataset. S.D. denotes Structural Distance. For A+B, A and B represent inversion and editing methods, respectively.

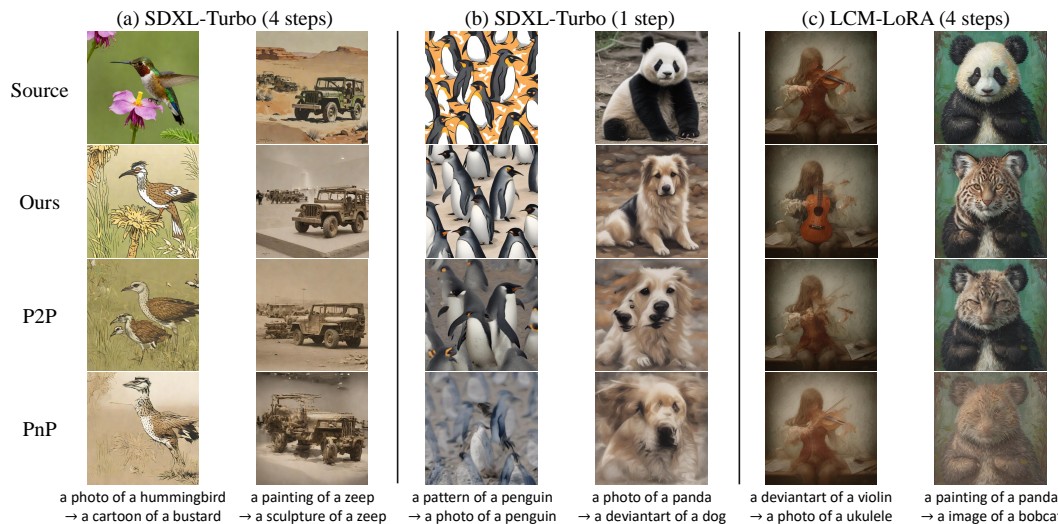

Figure 6: Editing results on ImageNet-R-TI2I dataset. The first row contains the source images, and the other rows show edited images by various editing techniques.

our method accurately edits the image to be matched with the text prompt while maintaining the structure of the source image in the generated image setting.

**Comparison to baselines on real images.** We also compare our method with baselines on the PIE-Bench dataset in real image settings. As shown in Fig. 7, our approach preserves the structural attributes of the real input image more effectively. In contrast, the baseline methods often lose the structure of the source image or fail to fully capture the context of the edit prompt. For example, in the 4[th] row, TurboEdit struggles to preserve the small bottle on the table, while EF (Huberman-Spiegelglas et al., 2024) and EF+P2P generate images similar to the source but fail to apply the 'Watercolor of' style. Notably, our method produces the edited output in only four steps, whereas EF requires 50 steps for the same task. Note that, TurboEdit here denotes TurboEdit[2] (Deutch et al., 2024), which is the only publicly available editing method in few-step DMs.

## 4.4 ABLATION STUDY

To validate the effectiveness of the proposed components, we conduct ablation studies presented in Tab. 3 using SDXL-Turbo (4-step) and Generated ImageNet-R-TI2I dataset. First, vanilla feature injection (Config A) successfully transfers the source structure, as shown in Fig. 9. However, this results in the synthesized images being overly constrained by the source structure, limiting their ability to be reliably edited. When we decompose structure and appearance information (Config B), the CLIP score improves significantly, indicating better alignment with the text prompt. Although this configuration reports a higher Structure Distance, it is important to note that this metric is not optimal for editing tasks, as some loss of structure due to attribute leakage is inevitable. Finally, Config C demonstrates that the proposed spatial matching improves structure transfer performance, albeit with a marginal decrease in the CLIP score.

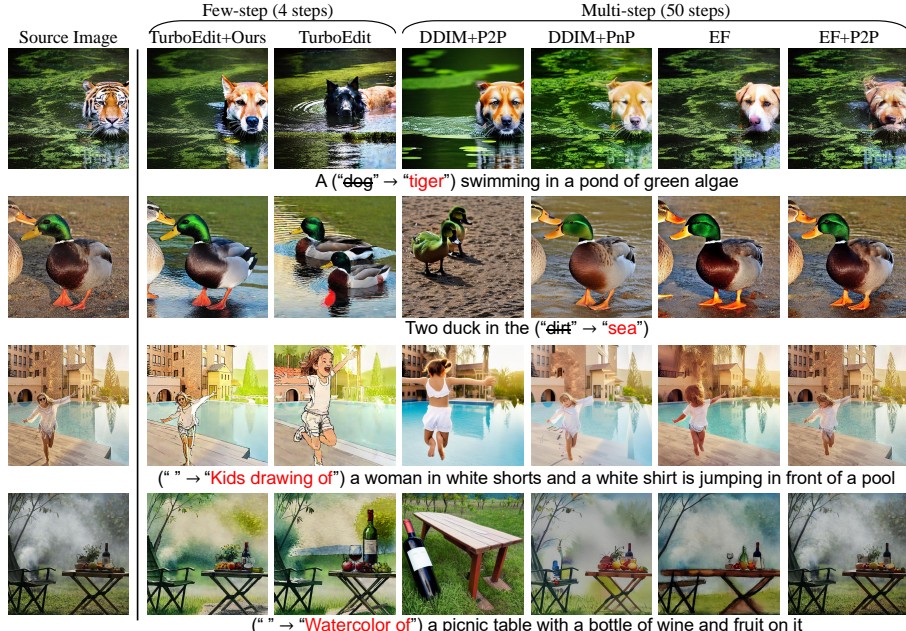

Figure 7: Editing results on PIE-Bench dataset. The first column contains the source images, and the other columns show edited images by various editing techniques.

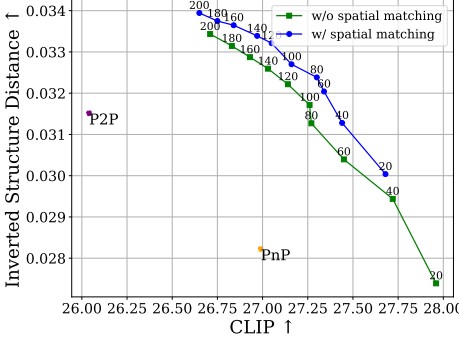

Figure 8: Performance trade-off while varying $k$ and ablating Spatial Matching.

|  | Configuration | CLIP ↑ | S.D. ($\times 10^3$) ↓ |
|---|---|---|---|
| A | Vanilla Feat. Inj. | 24.26 | 24.77 |
| B | + S-A Decomp. ($k = 60$) | **27.45** | 32.90 |
| B* | + S-A Decomp. ($k = 80$) | 27.27 | 31.98 |
| C | + Spatial Matching | 27.34 | **31.21** |

Table 3: Ablations on proposed components.

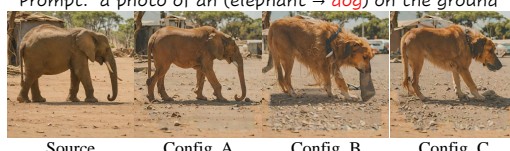

Figure 9: Qualitative results w/ ablations.

We also present a performance trade-off graph to assess the impact of spatial matching with varying values of K (20-200). As shown in Fig. 8, increasing K creates a trade-off between structure and appearance. However, the introduction of spatial matching mitigates this trade-off, as evidenced by improved CLIP and Inverted Structure Distance scores in the upper-right region of the graph.

## 5 CONCLUSION

This work addresses the limitations of attention-injection-based text-to-image (T2I) editing methods, particularly their inability to preserve the source image structure when applied to few-step diffusion models (DMs). Our analysis reveal that attention layers in few-step DMs have less influence on determining the spatial layout. To resolve this, we identify specific positions within the network that determine spatial layout and injected features at these points during the editing process. We also mitigate attribute leakage by disentangling structural factors using the singular values of source features. This decomposition isolates structural information, ensuring that the appearance attributes from text prompt do not conflict with those of the injected source features. Further, we address misalignment issue between source structure and target appearance factors with a warping-based spatial matching step, leading to improved visual quality. Extensive experiments demonstrate that our method effectively captures both the source structure and the text prompt, outperforming baseline methods in both quantitative and qualitative results.

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

# A  ABLATION STUDY ON VALUE OF K AND $\tau$

In our method, we determine the threshold index K of singular values for separating structural information from features. In this paragraph, we additionally conduct experiments to analyze the CLIP score and Structure Distance while varying the threshold index K. As shown in Fig. 10, a larger value of K increases the contribution of structural information after projection, enhancing the alignment of the projected features with the source structure. However, this also leads to a decrease in the CLIP score, highlighting the trade-off relationship between these two metrics. For all experiments, we use $k = 60$.

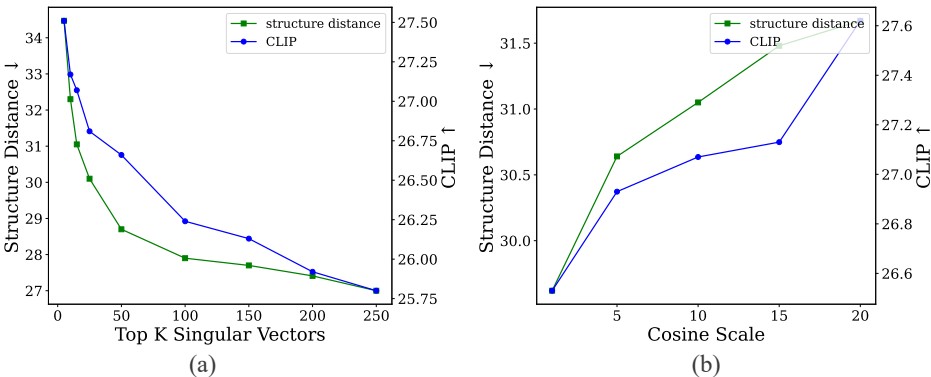

Figure 10: CLIP and structure distance while varying the threshold index K. (left) As the value of k increases, both the CLIP score and structure distance tend to decrease. (right) As $\tau$ increases, both the CLIP score and structure distance tend to increase.

# B  EXPERIMENTS ON LCM-LoRA

We perform the identical experiments in Fig. 3 and Fig. 4, using LCM-LoRA. As shown in Fig. 11-(a), the feature injection in latter blocks significantly low structural distance, indicating it synthesizes almost the identical source image. Also, the mid block (Block #6) shows the highest structural distance, as this layer has the most semantic information. Similarly, the output of the last encoder block (Block #5) has a low structural distance with a proper CLIP score, which is appropriate for the editing task.

We also visualize the spatial information of features from each block, in Fig. 11-(b). As shown, the structure distance rapidly decreases as K increases, while error toward the original features largely

remains. Thus, it implies that spatial information of images mostly resides in the high singular vectors, as same as the claim in the SDXL-Turbo.

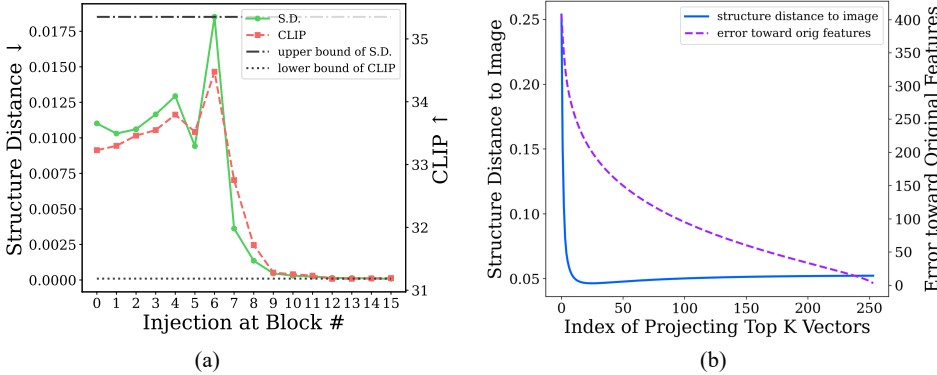

Figure 11: Analysis on LCM LoRA for feature injection.

## C  ANALYSIS OF RESIDUAL FEATURE INJECTION

To assess how much the residual feature affects the structure of the final output image, we injected the source residual feature from each ResNet block of the Diffusion model and calculated the structure similarity distance between the resulting images. As shown in Fig. 12, injecting the residual feature from Block#5 resulted in the lowest scores, indicating the best preservation of the source structure. However, the structure distance values are significantly higher compared to the structure distance observed with the non-residual feature injection at the Block#3 in our method, as depicted in Fig. 4. Through this, we confirm that non-residual features are more effective than residual features in preserving the structure.

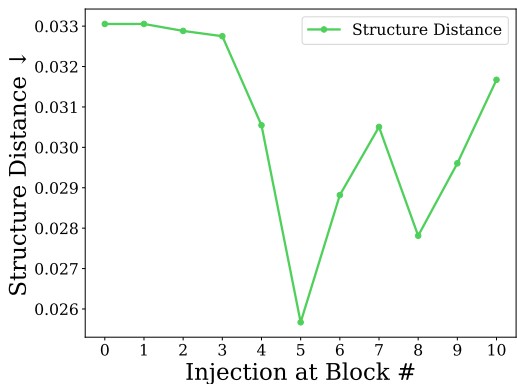

Figure 12: Analysis of Residual Feature Injection.

