# OpenReview forum: "Structure-Preserving Text-Based Editing for Few-Step Diffusion Models"
_ICLR.cc/2025/Conference — ICLR 2025 Conference Withdrawn Submission_

### Official Review · Reviewer_PPEv · 2024-10-23

**Soundness:** 2
**Presentation:** 3
**Contribution:** 2
**Rating:** 3
**Confidence:** 5

**Summary:**

This paper addresses the challenge of text-based image editing in few-step diffusion models, focusing on preserving the spatial structure of the original image while aligning it with a given text prompt. Existing methods often rely on attention maps for structure preservation, but these are less effective in faster few-step models like SDXL-Turbo and LCM. The authors propose a new approach that identifies and injects spatial layout features from specific layers in the model to better control structure. By disentangling structure from other features, their method avoids conflicts between the image and text, outperforming current baselines in both speed and accuracy.

**Strengths:**

1. This paper addresses a highly relevant and intriguing problem—image editing using few-step diffusion models. While previous research in this area is sparse, the authors tackle a significant and under-explored challenge.

2. The paper offers analysis of the unique characteristics of few-step models in the context of image editing. The proposed method is both simple and efficient, showcasing impressive speed. Additionally, the results appear compelling.

3. The paper is clearly written, with a well-organized structure, making it easy to follow and understand.

**Weaknesses:**

1. The technical contribution of this paper is somewhat limited.
- While the authors provide an analysis of the characteristics of few-step models, they do not offer a detailed explanation as to why these models exhibit such characteristics compared to non-few-step models.
- The technical novelty of the proposed editing method, SPEdit, is limited. Its primary contribution lies in the use of SVD for structure-appearance decomposition, while other parts of SPEdit heavily draw from previous works. For instance, the approach to spatial matching with a correspondence map closely resembles techniques used in DreamMatcher [1] and Consistory [2].

2. The experimental section is not comprehensive enough. The authors only compare their method with PnP (CVPR 2023) [3] and P2P (ICLR 2023) [4], which are relatively outdated baselines. There has been a recent surge in text-based image editing methods, such as FreePromptEditing (CVPR 2024) [5] and InfEdit (CVPR 2024) [6], whose performance on few-step models remains underexplored. These more recent methods should be included in the comparisons.

3. Limited application scenarios. The proposed method focuses on preserving the spatial layout of the edited image to match the original, which seems best suited for rigid image editing tasks (e.g., style changes). However, its effectiveness in non-rigid editing scenarios (e.g., raising a person's arm) remains unclear.

4. Generalization capabilities. The authors only test their method on models based on SDXL, such as SDXL-Turbo and LCM. It is unclear whether the proposed method can generalize to other few-step models like Flux or SD3. Given that models like Flux also support few-step inference and have strong generation capabilities, the authors should explore the applicability of their method to these models.

*_[1] DreamMatcher: Appearance Matching Self-Attention for Semantically-Consistent Text-to-Image Personalization. CVPR 2024._*

*_[2] ConsiStory: Training-Free Consistent Text-to-Image Generation. SIGGRAPH 2024._*

*_[3] Plug-and-Play Diffusion Features for Text-Driven Image-to-Image Translation. CVPR 2023._*

*_[4] Prompt-to-Prompt Image Editing with Cross Attention Control. ICLR 2023._*

*_[5] Towards Understanding Cross and Self-Attention in Stable Diffusion for Text-Guided Image Editing. CVPR 2024._*

*_[6] Inversion-Free Image Editing with Natural Language. CVPR 2024._*

**Questions:**

My main concern is outlined in the weaknesses section above. Here are some additional questions:

1. Could the authors provide a detailed analysis explaining why the effects of attention layers in few-step models are significantly lower compared to their role in multi-step diffusion models (as noted in Line 160)? Specifically, if a standard SDXL model is augmented with an LCM-LoRA, what mechanisms enable it to exhibit characteristics typical of few-step models?

2. It appears that the proposed SPEdit method struggles to maintain certain visual details, as evidenced by the noticeable differences in the duck's tail in the second row of Fig. 7. Is this discrepancy a result of the inversion process?

3. Can SPEdit be utilized for low-level editing tasks, such as adjusting lighting, removing haze, or erasing text from images? If so, how does the method adapt to these specific editing requirements? If not, what limitations prevent it from performing these types of edits, and are there plans to extend its functionality in this direction?

---

### Official Review · Reviewer_nQnr · 2024-10-29

**Soundness:** 2
**Presentation:** 2
**Contribution:** 2
**Rating:** 3
**Confidence:** 5

**Summary:**

The paper presents a novel approach for prompt-based image editing in diffusion models that minimizes editing steps. By leveraging feature injection rather than traditional attention map replacement, the proposed method preserves the structure of the original image. To improve accuracy in editing tasks, the paper introduces the use of Singular Value Decomposition (SVD) to separate appearance and structural information, facilitating selective modifications in appearance or structure. Additionally, the authors incorporate a spatial matching technique to further enhance the quality of the edited images.

**Strengths:**

- The paper is clearly written and accessible, enabling a solid understanding of the methodology.
- The proposed feature injection approach effectively maintains structural integrity in the edited images, representing an improvement over previous methods.

**Weaknesses:**

1. **Visual Inconsistencies in Results**: In Figure 1, the edited result introduces a lower right leg, whereas the original source image lacks this feature. This is an issue not present in the PnP results and raises concerns about the preservation of structural accuracy.
2. **Specification of Attributes**: Line 55 states, "Attributes not explicitly specified in the prompt should be preserved as in the original source image." However, the paper does not clarify how users can explicitly select attributes to be preserved or edited, which might limit practical applications.
3. **Clarity in Figures and Captions**: The caption for Figure 3 is somewhat ambiguous. It reads, “Structure distance between the source and the attention- or feature-injected images, and CLIP similarity between the source image and the text condition.” It would be clearer as, “CLIP similarity between the edited image and the text condition.” Additionally, specify in the figure whether it depicts structure distances for attention-injected or feature-injected images. Clearer explanations of settings or parameters used for the attention-injection process, if any, are also recommended.
4. **User Study Recommendation**: A user study would provide valuable insights into the practical impact of the proposed method on image quality and structural fidelity.
5. **Overemphasis on Structural Constraints**: The proposed method heavily constrains the edited image to maintain the structural layout of the original source image, which limits its flexibility in addressing more diverse editing tasks. By enforcing structural similarity between the original and target images, as well as the alignment between source and edited prompts, the approach may struggle with tasks that require more extensive transformations, such as object removal, significant pose adjustments, or arbitrary object replacements. This constraint introduces a potential gap in applicability for broader text-based image editing needs.

**Questions:**

1. **Handling Broader Editing Tasks**: How does the proposed method handle tasks such as object removal or pose alteration, and how does it perform on such tasks compared to other methods in the PIE Bench?
2. **Exploration of Feature Injection Across Blocks**: Have the authors experimented with using multiple blocks for feature injection instead of focusing on a single block? If so, what were the observed effects?
3. **Comparison with Alternative Methods**: Why were methods such as MasaCtrl [1] and FPE [2] not included in the comparative analysis?
4. **Discussion on Method Limitations**: The paper does not discuss the limitations of this approach. A discussion of limitations would enhance the completeness of the study.


#### References
1. MasaCtrl: Tuning-free Mutual Self-Attention Control for Consistent Image Synthesis and Editing.
2. Towards Understanding Cross and Self-Attention in Stable Diffusion for Text-Guided Image Editing. In *Proceedings of the IEEE/CVF Conference on Computer Vision and Pattern Recognition*, pp. 7817–7826, 2024.

---

### Official Review · Reviewer_K7Mi · 2024-10-30

**Soundness:** 2
**Presentation:** 3
**Contribution:** 2
**Rating:** 5
**Confidence:** 5

**Summary:**

This paper aims to address the challenge of structure preservation in few-step DMs for text-based image editing. There are three key designs: 1) Instead of injecting the SA and CA maps from the source to the target generation process, this paper injects the features in the non-residual path to perserve more structure information. 2) The SVD technique is applied to filter out the structure-irrelevant information from the features, retaining only the top-k components. 3) This paper employs spatial matching technique to further improve the editing quality.

**Strengths:**

1) The paper is well-written and provides the necessary analysis to support its claims.
2)  Injecting non-residual features from the source to the target generation process is novel and demonstrates some effectiveness.
3) The proposed method achieves SOTA performance on two benchmark datasets.

**Weaknesses:**

1. Lacking comprehensive comparison with the latest methods. The methods being compared in this paper, P2P (ICLR 2023) and PnP (CVPR 2023), are nearly two years old.
2. The reliability of extracting structure-related features using the top-k significant singular vectors (Sec. 3.2) seems questionable. Would this approach remain effective for arbitrary images? For instance, how well would it perform on an image where the foreground object occupies nearly the entire area? In such a case, the significant singular vectors would primarily capture the object's appearance.
3. Lacking quantitative experiments to assess the effectiveness of filtering out appearance factors.

**Questions:**

1. Both few-step and multi-step DMs utilize similar residual connections within the U-Net architecture, why the output residual features of the self-attention (SA) and cross-attention (CA) blocks in few-step DMs are easier to be ignored and results in less contributions in determining the spatial layout.
2. Why not inject appearance features from the target into the source generation process, as done in P2P? It seems easier in preserving structure (See the examples in the project page of P2P: https://prompt-to-prompt.github.io/).
3. Which dataset did you use for the experiments shown in Fig. 4(a)?
4. Could you provide more results about modifying only part of the image while leaving other regions unchanged?

I look forward to your response demonstrating the necessity of your proposed method and its effectiveness on arbitrary images. I will revise my score based on your answers and feedback from other reviewers.

---

### Official Review · Reviewer_EitT · 2024-11-04

**Soundness:** 3
**Presentation:** 3
**Contribution:** 3
**Rating:** 5
**Confidence:** 4

**Summary:**

The paper proposes feature injection method for structure preserved attribute editing.

**Strengths:**

The method is simple and outperforms previous methods.

**Weaknesses:**

1. Although the method is simple and effective, the comparison between other methods is still not enough. the paper mainly compare their results with PNP and P2P and some of TurboEdit. There are so many editing methods which use fast diffusion models. Please add more comparison results.

2. The evaluation metric of CLIP score and structural distance is not enough. As editing is mainly on the subjective field, I recommend to add User study not just using automatic metrics.

3. The method has disadvantage on its instability. We have to find a appropriate 'sweetspot' of layer for different models. also the method might not be adapted to other kinds of generative models such as Diffusion Transformer.

4. Is this method still be able to be applied on non-rigid editing? for example, making a dog into 'jumping dog'. Structure-preserved image translation is not a difficult task and there are so many alternative methods.

5. In baseline method of PNP and P2P, users can control the structure preservation degree with changing the timestep of attention injection. I think the degraded structural preservation comes from this. Please compare the results with various attention injection timesteps.

**Questions:**

See weakness

---

> ### Comment · Reviewer_EitT · 2024-12-02
>
> As the authors did not upload any rebuttals, I will keep my original rating.

---

### Note · Authors · 2025-01-17

I have read and agree with the venue's withdrawal policy on behalf of myself and my co-authors.